# Characteristics of Skin Deposition of Itraconazole Solubilized in Cream Formulation

**DOI:** 10.3390/pharmaceutics11040195

**Published:** 2019-04-22

**Authors:** Hyeongmin Kim, Sukkyun Jung, Sooho Yeo, Dohyun Kim, Young Chae Na, Gyiae Yun, Jaehwi Lee

**Affiliations:** 1College of Pharmacy, Chung-Ang University, Seoul 06974, Korea; hm.kim8905@gmail.com (H.K.); jsk0314@cau.ac.kr (S.J.); sooho32@hanmail.net (S.Y.); dylan13@naver.com (D.K.); nyc920916@naver.com (Y.C.N.); 2Department of Food Science and Technology, Chung-Ang University, Anseong 17546, Korea; gyiae@cau.ac.kr

**Keywords:** itraconazole, cream formulation, solubilization, skin deposition, cutaneous mycoses

## Abstract

Itraconazole (ITZ) is an anti-fungal agent generally used to treat cutaneous mycoses. For efficient delivery of ITZ to the skin tissues, an oil-in-water (O/W) cream formulation was developed. The O/W cream base was designed based on the solubility measurement of ITZ in various excipients. A physical mixture of the O/W cream base and ITZ was also prepared as a control formulation to evaluate the effects of the solubilized state of ITZ in cream base on the in vitro skin deposition behavior of ITZ. Polarized light microscopy and differential scanning calorimetry demonstrated that ITZ was fully solubilized in the O/W cream formulation. The O/W cream formulation exhibited considerably enhanced deposition of ITZ in the stratum corneum, epidermis, and dermis compared with that of the physical mixture, largely owing to its high solubilization capacity for ITZ. Therefore, the O/W cream formulation of ITZ developed in this study is promising for the treatment of cutaneous mycoses caused by fungi such as dermatophytes and yeasts.

## 1. Introduction

Itraconazole (ITZ) is a triazole structure-based anti-fungal agent that is generally used to treat cutaneous mycoses owing to its high activity against a broad spectrum of pathogenic fungi causing the disease, such as dermatophytes and yeasts [1]. In clinical practice, ITZ is orally administered with a typical dose of 200–400 mg/day to treat cutaneous mycoses [2,3]. However, systemic exposure to ITZ resulting from oral administration has been frequently reported to cause liver damage due to hepatocellular and cholestatic damage [4,5]. Besides the avoidance of adverse effects of ITZ on the liver, the drug should be delivered to the skin tissues, specifically to the stratum corneum, which is the outermost layer of the skin and the main target site for the treatment of cutaneous mycoses [6]. Therefore, topical delivery systems for efficient delivery of ITZ to the stratum corneum are needed. Despite this, the topical dosage forms of ITZ are rarely available on the market due largely to its extremely insoluble nature with aqueous solubility reported to be 1 ng/mL [7]. 

Numerous topical dosage forms of poorly soluble drugs have been investigated, such as liposomes, solid lipid nanoparticles, and nanostructured lipid carriers [8,9,10,11]. These are particulate or colloidal carrier systems, which are advantageous to some extent in solubilizing poorly soluble drugs, but exhibit limitations in delivering the drugs to the skin tissues such as the stratum corneum and deeper tissues (i.e., epidermis and dermis). The particulate carrier systems are transported mainly through hair follicles or sweat glands, and thus the delivery efficiency is poor because the total fraction of these skin appendages is small [12]. In addition, the drug incorporated in the particulate systems should be released in order to be effective, but the drug release rate is not sufficient to initiate rapid action of the drug. 

Microemulsions and nanoemulsions are also considered promising topical formulations for efficient delivery of poorly soluble drugs such as ITZ to the stratum corneum and deeper skin tissues. This is because they offer solubilized ITZ and thereby the drugs can be rapidly absorbed into the skin tissues through the lipophilic domain of the stratum corneum. However, they possess free-flowing nature, and thus, the residence time of the drug on the skin is very short. Therefore, microemulsions and nanoemulsions have been incorporated into the hydrogel formulations (i.e., microemulsion-/nanoemulsion-based hydrogels) to increase the residence time on the skin, but the drug-release rate is severely reduced due to increased diffusion barrier properties caused by the polymer chain networks [13,14,15,16]. Another disadvantage associated with microemulsions and nanoemulsions is low drug-loading capability.

To overcome the limitations and disadvantages related to the current topical formulations applicable for ITZ, we explored oil-in-water (O/W) type cream formulations for the topical delivery of ITZ to the skin, specifically into the stratum corneum. The O/W creams consist of oil droplets dispersed in water, and they exhibit a viscous texture advantageous to reside longer on the skin. If the oil droplets can solubilize ITZ efficiently, the drug would permeate the stratum corneum and the deeper skin tissues such as the epidermis. Therefore, the present study aimed to design cream formulations and to evaluate the skin tissue distribution profiles of ITZ, thereby improving drug action and providing topical dosage forms of ITZ.

## 2. Materials and Methods

### 2.1. Materials

ITZ, cetyl alcohol, and cetostearyl alcohol were purchased from Sigma-Aldrich Company (St. Louis, MO, USA). Labrafac^®^ CC (caprylic/capric triglyceride) and glyceryl monostearate were obtained from Gattefosse (Lyon, France). Tween^®^ 80 (polyoxyethylene sorbitan monooleate), Tween^®^ 40 (polyoxyethylene sorbitan monopalmitate), Span^®^ 60 (sorbitan monostearate), Span^®^ 80 (sorbitan monooleate), stearic acid, paraffin oil, mineral oil, and propylene glycol were purchased from Duksan Pure Chemicals Co., Ltd. (Seoul, Korea). Polyglyceryl-3 methylglucose distearate was procured from Goldschmidt (Essen, Germany). Acetonitrile and methanol of high-performance liquid chromatography (HPLC) grade were purchased from J.T. Baker Chemical Company (Phillipsburg, NJ, USA). Distilled and deionized water was used to prepare all the solutions.

### 2.2. Solubility Measurement of ITZ in Excipients for Cream Formulations 

The solubility of ITZ in different excipients (oils, emulsifiers, glycerol esters of fatty acid, fatty acid, and fatty alcohols) (Table 1) that can be used to prepare cream formulations was evaluated. The excipients were heated up to 80 °C to melt them and an excess amount of ITZ was added to the excipients in a liquid state and stirred. The mixtures were maintained under the high-temperature condition for 1 h. The mixtures were then cooled to room temperature (~22 °C) and filtered through a 0.45-μm nylon filter to separate the undissolved ITZ. The filtrates were appropriately diluted with methanol, and the levels of ITZ in the diluted filtrates were assessed by HPLC. The solubility of ITZ in the excipients was also assessed at 20 °C if the excipients were in a liquid state under the temperature condition.

### 2.3. HPLC of ITZ 

The levels of ITZ in the samples were evaluated using the Breeze™ 2 HPLC system (Waters, Milford, MA, USA) equipped with the Zorbax^®^ C_18_ column (150 mm × 4.6 mm, 5-µm particle size; Agilent Technologies, Wilmington, DE, USA). The mobile phase was composed of acetonitrile and 20 mM potassium phosphate buffer (pH 10) at a ratio of 60:40 (*v*/*v*). The flow rate was set at 1.2 mL/min, and the UV detection wavelength was 254 nm. The column temperature was maintained at 30 °C.

### 2.4. Preparation of O/W Cream Formulation and Physical Mixture

Table 2 shows the composition of the O/W cream formulation of ITZ established based on the result of the solubility test of ITZ in different excipients. The oil phase consisting of Labrafac^®^ CC, Tween^®^ 80, glyceryl monostearate, stearic acid, and cetyl alcohol was heated up to 80 °C, and then ITZ was added to the oil phase and mixed by stirring for 10 min. After completely dissolving the drug in the oil phase, the aqueous phase composed of propylene glycol and water was added to the oil phase and heated at 80 °C. The mixture was homogenized for 30 min and degassed under vacuum condition. The O/W cream formulation was then cooled to room temperature (~22 °C). 

To prepare the physical mixture of ITZ and the O/W cream base (physical mixture) as a control formulation, the oil phase devoid of ITZ and the aqueous phase were heated up to 80 °C, respectively. The oil phase was added to the aqueous phase, homogenized for 30 min at the same temperature, and cooled to room temperature (~22 °C). ITZ raw material powder was then added to the cream base and stirred to obtain the physical mixture. The level of ITZ in the O/W cream formulation and the physical mixture was set at 1% (*w*/*w*). 

### 2.5. Solubility Measurement of ITZ in Oil Phase and O/W Cream Base 

An excess amount of ITZ was added to the oil phase composed of different excipients as shown in Table 2. The mixture was stirred at 20 °C or 80 °C for 1 h. The mixture was then cooled to room temperature (~22 °C) and filtered through a 0.45-μm nylon filter to separate the undissolved ITZ. The filtrate was diluted with methanol, and the level of ITZ in the diluted filtrate was evaluated by HPLC. 

As for the solubility measurement of ITZ in the O/W cream base, the O/W cream base was heated at 60 °C to be melted with maintaining the emulsified state of the cream base. An excess amount of ITZ was added to the melted cream base and stirred for 1 h at 60 °C. The mixture of the melted cream base and ITZ was cooled to room temperature (~22 °C), following the filtration of the mixture with a 0.45-μm nylon filter to separate the undissolved drug. The filtrate was diluted with methanol, and the level of ITZ in the diluted filtrate was assessed by HPLC.

### 2.6. Characterization of O/W Cream Formulation and Physical Mixture

#### 2.6.1. Organoleptic Properties

The cream formulation and the physical mixture were visually examined for color, texture, and phase separation. The feel of the cream formulation and physical mixture such as stiffness, grittiness, greasiness, and irritation was also examined. Written informed consent was obtained after thoroughly explaining the study including purpose and risks. The study was conducted under the ethical guideline of the Declaration of Helsinki (version 2000; World Medical Association). In accordance with the review guideline set by the Institutional Review Board of Chung-Ang University, a review was not necessary since the study used cream formulations prepared using ingredients that met the safety guideline of Korean Cosmetic Low Article 8 and did not involve invasive procedures. Ten human volunteers aged between 24 and 30 applied approximately 150 mg of the O/W cream formulation and the physical mixture on each back of their hands using a spatula and gently rubbed the cream formulation and physical mixture. After 2 min, the participants evaluated the feel of the cream formulation and physical mixture applied to their skin such as stiffness, grittiness, greasiness, and irritation. A small amount of each cream formulation and the physical mixture was also pressed using the thumb and index finger to check the presence or absence of solid drug powder. 

#### 2.6.2. pH

The pH of the O/W cream formulation and the physical mixture was evaluated using a pH paper (Fisherbrand™ Paper pH Strips, Thermo Fisher Scientific Inc., Waltham, MA, USA). A small quantity of the cream formulation and the physical mixture was applied on the one side of the pH paper. The pH of the cream formulation and the physical mixture was determined by comparing the change in color of the pH paper on the other side with the color chart.

#### 2.6.3. Polarized Light Microscopy

The O/W cream base devoid of ITZ, O/W cream formulation in which ITZ was dissolved, and the physical mixture were examined by using a polarized light microscope (Olympus BX51, Olympus Optical Co. Ltd., Tokyo, Japan). An appropriate amount of the samples was placed on a glass slide. The samples were observed under crossed polars at 100× magnification.

### 2.7. Differential Scanning Calorimetry

Thermal characteristics of ITZ, the oil phase of the O/W cream formulation devoid of the drug, and the oil phase in which the drug was dissolved were assessed by differential scanning calorimetry (DSC) (DSC822e; Mettler-Toledo, Columbus, OH, USA). An appropriate amount of the samples was sealed in an aluminum crimp cell and heated from 25 °C to 200 °C at a rate of 10 °C/min under nitrogen atmosphere. The flow rate was set at 30 mL/min in the temperature range.

### 2.8. Stability Test of O/W Cream Formulation of ITZ 

Immediately after preparing the O/W cream formulation, the content of ITZ in the O/W cream formulation was assessed. The cream formulation was placed in 20 mL glass vials and sealed with paraffin film. Subsequently, the cream formulation was stored under two different temperature conditions (25 °C and 40 °C) for 8 weeks. The changes in the drug content in the cream formulation were evaluated at 4 weeks and 8 weeks. To assess the drug content in the cream formulation, an appropriate quantity of the cream formulation was weighed, diluted with methanol, and then subjected to HPLC for quantification of the drug. In addition to the drug content, the pH and properties of the cream formulation such as color, texture, and phase separation were also examined simultaneously. 

### 2.9. In Vitro Skin Deposition and Penetration Analyses

#### 2.9.1. Preparation of Rat Skin

The protocol of animal care and use was reviewed and approved by the Institutional Animal Care and Use Committee at Chung-Ang University (approval number: 2017-00002; approval date: 10 January 2017). Male Wistar rats aged 6–8 weeks were obtained from the Hanlim Experimental Animals Co., Ltd. (Hwasung, Korea). The rats were epilated on the dorsal side using an animal hair clipper and shaving razor in the direction of the tail to head without damaging the skin tissues. The skin was sampled with the rats under deep surgical anesthesia and the subcutaneous lipid was removed. The rats were euthanized using bicarbonate gas, and all efforts were made to minimize suffering to the rats. The excised skin was rinsed with PBS and used for the in vitro skin permeation and deposition analyses. 

#### 2.9.2. In Vitro Skin Penetration Analysis

The freshly excised rat skin samples were mounted between the donor and receiver compartments of Franz diffusion cells with the stratum corneum of the rat skin facing the donor compartment. Each Franz diffusion cell had a diffusional surface area of 1.76 cm^2^ and the volume of the receiver compartment was 11.0 mL. The temperature of the receiver compartment was maintained at 36.5 °C using an external constant-temperature circulator water bath. The receiver compartment was filled with 1 M *N*,*N*-diethyl nicotinamide (DENA) in phosphate buffer (pH 7.4), which was used to maintain sink condition in the receiver cell and agitated by magnetic stirring. The O/W cream formulation and physical mixture containing ITZ equivalent to 6.5 mg of the drug were applied on the membrane in the donor compartment, ensuring an intimate contact with the rat skin. The donor compartment was covered with paraffin film to prevent the evaporation of water contained in the cream formulation and physical mixture. At predetermined time intervals (6, 12, and 24 h), aliquots (0.5 mL) were sampled from the sampling arm of the receiver compartment, and the same volume of the fresh receptor medium was added to the receiver compartment. The level of ITZ in the aliquots was determined by HPLC.

#### 2.9.3. In Vitro Skin Deposition Analysis

To evaluate the deposition of ITZ in the rat skin tissues, at the end of the skin permeation experiment, the surface of the rat skin was rinsed with distilled water and gently wiped with tissue paper to remove residual formulations. To separate the stratum corneum, the rat skin samples were stripped six times with an adhesive tape (Transpore™, 3M; St. Paul, MN, USA). The weight of the stratum corneum separated by the tape-stripping method was determined by the difference in weight of the adhesive tapes before and after stripping. The adhesive tapes were then immersed in methanol, vortexed for 10 min, sonicated for 60 min, and shaken in a horizontal shaker for 5 h to dissolve ITZ contained in the stratum corneum attached to the adhesive tapes. The methanolic solution was filtered through a 0.45-μm nylon filter and the level of ITZ in the filtrate was assessed by HPLC.

After tape stripping, the epidermis of the rat skin samples was separated from the dermis by heat treatment. The skin samples were immersed in distilled water maintained at 60 °C for 1 min and the epidermis was carefully removed from the dermis using a pair of forceps. The separated epidermis and dermis were then cut into small pieces and immersed in methanol. The epidermis and dermis in methanol were homogenized for 5 min and shaken in a horizontal shaker for 5 h. The resulting homogenates were then filtrated using 0.45-μm nylon filters and the level of ITZ in the filtrates was determined by HPLC.

### 2.10. Statistical Analysis

All experiments were conducted in quadruplicate. Means were compared by one-way analysis of variance and Student’s *t*-test. *P* < 0.05 was considered significant.

## 3. Results and Discussion

### 3.1. Solubility of ITZ in Excipients and O/W Cream Formulation 

The aqueous solubility of ITZ has been known to be extremely low, approximately 1 ng/mL [7]. Because of the low aqueous solubility of ITZ, it is challenging to dissolve the drug in topical formulations such as cream or ointment at therapeutically effective levels. To overcome this limitation, in the present study, we aimed to develop an O/W cream formulation with a maximized solubilization capacity for ITZ. For this purpose, the solubility of ITZ in various excipients such as oils, emulsifiers, glycerol esters of fatty acid, fatty alcohols, and fatty acid used to prepare cream formulations was assessed at 20 °C and 80 °C. The reason for evaluating the solubility of ITZ in the excipients at the high temperature (80 °C) was because it was necessary to melt thickeners such as glycerol esters of fatty acid, fatty alcohols, and fatty acid during the emulsification procedure based on the melting points of the thickeners. 

The solubility of ITZ evaluated in the excipients is shown in Table 1. In general, the solubility values of ITZ in the excipients at 80 °C were considerably greater than those measured at 20 °C. This result might be because the increase in kinetic energy under the high temperature (80 °C) allowed the excipient molecules to break apart the drug molecules more effectively than at 20 °C. Except for Labrafac^®^ CC, the oils tested were not able to solubilize ITZ. This might be due to the extremely hydrophobic nature of ITZ [17]. In case of some excipients, they were not fully liquid state at 20 °C so that the solubility of ITZ could not be assessed.

Based on the solubilities of ITZ demonstrated in Table 1, Labrafac^®^ CC as oil and Tween^®^ 80 as emulsifier were selected to form an O/W emulsion with the aqueous phase. Glyceryl mono-stearate, stearic acid, and cetyl alcohol were also added to the oil phase to finally produce O/W cream with a suitable texture. Numerous formulation works were done to optimize the cream formulation having a maximal solubilizing capacity of ITZ (data not shown). The composition of the optimized cream formulation for solubilizing ITZ is shown in Table 2, and the oil phase of the selected cream formulation solubilized approximately 18 mg/mL and 33 mg/mL of ITZ at 20 °C and 80 °C, respectively. The solubility of ITZ in the final O/W cream formulation was determined to be 1.5% (*w*/*w*) at room temperature (~22 °C). However, for further evaluation of the cream formulation, the content of ITZ in the O/W cream was set at 1% (*w*/*w*). 

### 3.2. Organoleptic Characteristics of O/W Cream Formulation and Physical Mixture

The O/W cream formulation of ITZ was white and opaque, with a smooth and semi-solid texture. Any bleeding and phase separation were not observed from the cream formulation. The O/W cream formulation did not contain any particles, indicating that ITZ was completely dissolved in the cream formulation. The O/W cream formulation was also easily spread on the skin. Two minutes after applying the cream formulation on the skin, the cream formulation did not show any undesirable textures such as tackiness and greasiness; it could be easily washed with water.

In the physical mixture, the drug powder was not dissolved; it was only physically dispersed in the O/W cream base. Although the physical mixture was easily spread on the skin, the drug powder remained on the skin, causing uncomfortable feeling and even physical irritation. 

### 3.3. Polarized Light Microscopy

The O/W cream formulation of ITZ and the physical mixture were observed using a polarized light microscope in comparison to the cream base devoid of the drug. As presented in Figure 1A, oil globules were observed to be finely dispersed in the aqueous phase of the cream base (Figure 1A). The O/W cream formulation of ITZ was similar to the cream base without the drug (Figure 1B). Thus, it was demonstrated that ITZ was successfully dissolved in the oil phase. However, the physical mixture exhibited crystalline drug particles distributed in the outer aqueous phase (Figure 1C), denoting that most of the drug added could not be dissolved in the cream base without the heating procedure used for the preparation of the O/W formulation of ITZ. 

### 3.4. DSC Analysis 

Figure 2 shows the DSC thermograms of the ITZ raw material powder, the oil phase of the O/W cream base devoid of ITZ, and oil phase in which the drug was dissolved at a concentration of 1% (*w*/*w*). The ITZ powder exhibited a sharp endothermic peak at 166 °C corresponding to the melting point of the crystalline drug. The oil phase of the cream base alone exhibited a broad endothermic peak at a temperature range of 30–40 °C. As for the oil phase with ITZ, the characteristic endothermic peak of ITZ at 166 °C was not observed, indicating that the drug was completely dissolved in the oil phase. The results demonstrated that the heating oil phase at 80 °C led to a fully dissolved state of ITZ in the oil phase, which was well maintained after cooling the oil phase to room temperature. 

### 3.5. Stability of the Cream Formulation of ITZ

The result of the stability test of the O/W cream formulation of ITZ is illustrated in Figure 3. The concentration of ITZ in the O/W cream formulation was not considerably changed at 25 °C and 40 °C during the 8-week test period. No appreciable phase separation was observed from the O/W cream formulation of ITZ throughout the test period. This implies that ITZ solubilized in the O/W cream formulation was stable and the dispersion stability of the O/W cream formulation was appropriately retained for the whole tested period. The pH of the O/W cream formulation of ITZ remained unchanged (pH 6.0) under the test temperature condition for 8 weeks. The organoleptic characteristics and texture of the O/W cream formulation also did not change significantly during the test period. Thus, the O/W cream formulation of ITZ was found to be stable under the experimental conditions.

### 3.6. In Vitro Skin Deposition and Penetration Analyses

For effective treatment of cutaneous mycoses, anti-fungal drugs such as ITZ should be delivered to the skin tissues, particularly to the stratum corneum, because dermatophytes, such as *Trichophyton*, *Epidermophyton*, and *Microsporum* spp., require keratins for their survival and growth and therefore, they colonize only keratinized tissues such as the stratum corneum [18]. In the case of cutaneous mycoses caused by yeasts such as *Cryptococcus neoformans* and *Candida albicans*, anti-fungal drugs should permeate into non-keratinized skin tissues such as the epidermis and dermis to achieve successful therapeutic outcomes [19,20]. Therefore, we assessed the in vitro skin permeation and deposition behavior of ITZ using the O/W cream formulation and physical mixture. As the primary aim of this study was to deliver ITZ efficiently to the skin tissues using the O/W cream formulation with a maximized solubilization capacity for the drug, the physical mixture in which the drug was supposed to be not soluble or only partially soluble was used as a negative control formulation. 

For the in vitro skin permeation and deposition analyses, rat skin was used owing to its advantages such as low variation in permeability, easy handling, and low cost [21]. Although drug permeation rates through human skin are generally slower than those through rat skin because of the difference between thicknesses of human and rat skins, it has been known that drug permeation behaviors assessed with human and rat skins are well correlated [21]. For this reason, rat skin was used to predict the permeation behavior of ITZ incorporated in the cream formulation in human skin. 

Figure 4 shows the deposition profiles of ITZ in the stratum corneum, epidermis and dermis assessed for 24 h after the start of the in vitro skin deposition analysis performed with the O/W cream formulation and physical mixture. In general, the concentration and amount of ITZ deposited in the skin tissues gradually increased with time, and they were the highest in the stratum corneum, followed by those in the dermis and epidermis. This implies that the drug formulated in the cream formulation and physical mixture was initially delivered to the outermost skin tissue, i.e., the stratum corneum and then diffused into deeper skin tissues such as the epidermis and dermis with time. In addition, keratins abundantly present in the stratum corneum are known to have high binding affinity to lipophilic drugs such as ITZ, and thereby they act as drug reservoirs [22,23,24,25,26]. For this reason, most of ITZ deposited in the stratum corneum were supposed to be bound to keratins and only remaining drugs were diffused into deeper skin tissues, resulting in considerably higher concentration and amount of the drug in the stratum corneum than those in the epidermis and dermis.

The result of the in vitro skin deposition analysis also revealed that the concentration of ITZ deposited in the stratum corneum was considerably higher than those deposited in the epidermis and dermis for the test period, whereas in general the amount of ITZ deposited in the stratum corneum was not significantly higher than those deposited in the epidermis and dermis. This might be caused by the difference in tissue weight among the skin layers. The weight of the stratum corneum obtained from the rat skin was measured to be approximately 11 times and 5 times lower than those of the epidermis and dermis, respectively. Thus, the amount of ITZ deposited in the stratum corneum was comparatively low despite the high concentration of ITZ deposited in the stratum corneum.

The O/W cream formulation exhibited largely higher concentrations and amounts of ITZ deposited in the stratum corneum than those of the physical mixture. As for the physical mixture, ITZ was not appropriately solubilized; it was present in the cream base as a solid powder. The dissolution of the poorly water-soluble drug and its permeation into the stratum corneum are thus highly limited. In contrast, ITZ was fully dissolved in the oil phase of the O/W cream formulation, and the oil droplets were supposed to be intimately contacted with the lipophilic domain of the stratum corneum. This was considered ultimately to cause the supply of solubilized ITZ to the stratum corneum, which could be useful for the treatment of cutaneous mycoses caused by dermatophytes that colonize only the stratum corneum. By the enhanced partition of ITZ to the stratum corneum, the diffusion of the drug into the deeper skin tissues might be facilitated.

The concentration and amount of ITZ deposited in the stratum corneum assessed with the O/W cream formulation were not largely varied during the test period, whereas they increased gradually with time in case of the physical mixture. This might be attributed to the limited binding capacity of keratins in the stratum corneum. When the binding sites of keratins were saturated with ITZ delivered by the O/W cream, the unbound ITZ was considered to permeate into deeper skin tissues such as the epidermis and dermis. Because of this, the concentration and amount of ITZ in the epidermis and dermis assessed with the O/W cream formulation increased at considerably faster rates than the physical mixture as shown in Figure 5. Therefore, it was demonstrated that the O/W cream formulation with a maximized solubilization capacity for ITZ was useful for efficient delivery of ITZ to the skin tissues such as the stratum corneum, epidermis, and dermis, which is required for effective treatment of cutaneous mycoses caused by dermatophytes and yeasts that colonize the skin tissues.

However, it should also be noted that rat skin exhibits different barrier properties from human skin. The thickness, degree of hydration, and lipid composition of the rat SC have been known to be considerably different from those of human [27]. Therefore, rat skin certainly has a limitation in predicting the deposition behavior of ITZ in human skin. For better simulation of the drug distribution in human skin, to date, various skin models have been investigated such as excised human skin obtained from cadavers or plastic surgery [27], porcine skin [28], snakeskin [29,30], and reconstructed human skin tissues [31,32]. Although the excised human skin is considered to be the best ex vivo model for predicting the drug deposition behavior in human skin, the use of excised human skin is not generally feasible during the initial development of novel topical dosage forms such as the O/W cream formulation designed in this study. Among animal skin models, porcine skin is known to be particularly similar to human skin in terms of the thickness [33], main lipid classes [34], and lamellar organization of the SC [35]. Reconstructed full human skin models are also considered to be promising for evaluating the skin absorption of topically applied drugs because of their structure with the desired composition and good reproducibility [27]. However, such skin models also have their own disadvantages such as different lateral packing of lipids in the SC and lack of the skin appendages [27]. Therefore, the delivery property of ITZ incorporated in the O/W cream formulation needs to be critically studied in different skin models to maximize the correlation of the drug distribution behaviors between the skin models and human skin.

From the in vitro skin permeation study, we found that ITZ was not present in the receiver medium, indicating that the drug solubilized in the cream formulations did not penetrate the rat skin. As aforementioned, keratins with a high binding affinity to ITZ might act as a drug reservoir in the stratum corneum [36]. Thus, a significant amount of ITZ deposited in the stratum corneum was supposed to be bound to keratins. Thereby, the remaining drug molecules in the stratum corneum were probably insufficient to provide a large concentration gradient of the drug among the skin tissues to promote the permeation of the drug into the rat skin.

## 4. Conclusions

The O/W cream formulation with a high solubilization capacity for ITZ was successfully developed based on the solubility measurement of various excipients that can be used to prepare cream formulations. The O/W cream formulation of ITZ was demonstrated to be able to deliver the drug more efficiently to skin tissues such as the stratum corneum, epidermis, and dermis compared with the physical mixture, owing to its high solubilization capacity for the drug. Therefore, the O/W cream formulation of ITZ is promising for the treatment of cutaneous mycoses caused by dermatophytes and yeasts that colonize the stratum corneum and deeper tissues such as the epidermis and dermis, respectively.

## Figures and Tables

**Figure 1 pharmaceutics-11-00195-f001:**
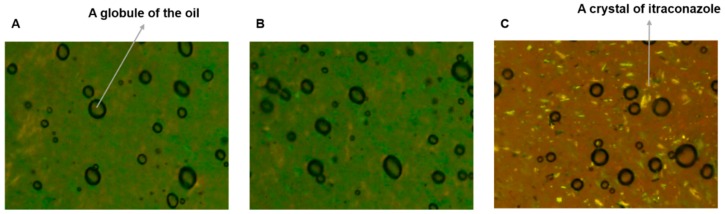
Images of the (**A**) O/W cream base devoid of ITZ, (**B**) O/W cream formulation of ITZ, and (**C**) physical mixture observed using a polarized light microscope at 100× magnification.

**Figure 2 pharmaceutics-11-00195-f002:**
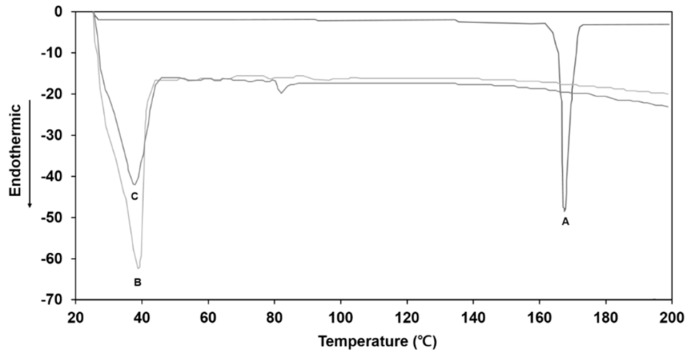
The DSC curves of (**A**) ITZ raw material powder, (**B**) oil phase of O/W cream base devoid of ITZ, and (**C**) oil phase in which ITZ was dissolved at a concentration of 1% (*w*/*w*).

**Figure 3 pharmaceutics-11-00195-f003:**
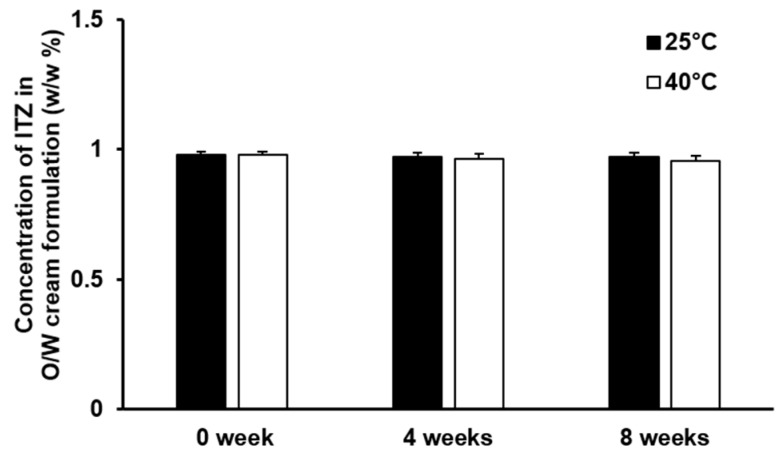
Stability of ITZ incorporated in the O/W cream formulation evaluated for 8 weeks at 25 °C and 40 °C. The values are presented as mean ± SD (*n* = 4).

**Figure 4 pharmaceutics-11-00195-f004:**
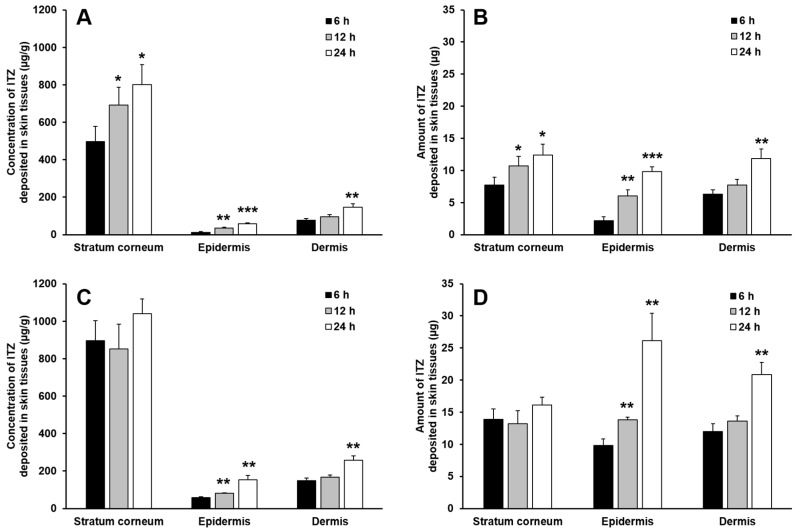
The deposited concentration of ITZ per gram tissue (**A**,**C**) and deposited amount (**B**,**D**) of ITZ in the stratum corneum, epidermis, and dermis evaluated with the physical mixture **(A**,**B**) and O/W cream formulation (**C**,**D**). The data are expressed as the mean ± SD (*n* = 4). Single, double, and triple asterisks denote statistical differences at *P* < 0.05, *P* < 0.01, and *P* < 0.001, respectively, between the concentration and amount of ITZ deposited in each skin tissue at 6 h and those at 12 h and 24 h.

**Figure 5 pharmaceutics-11-00195-f005:**
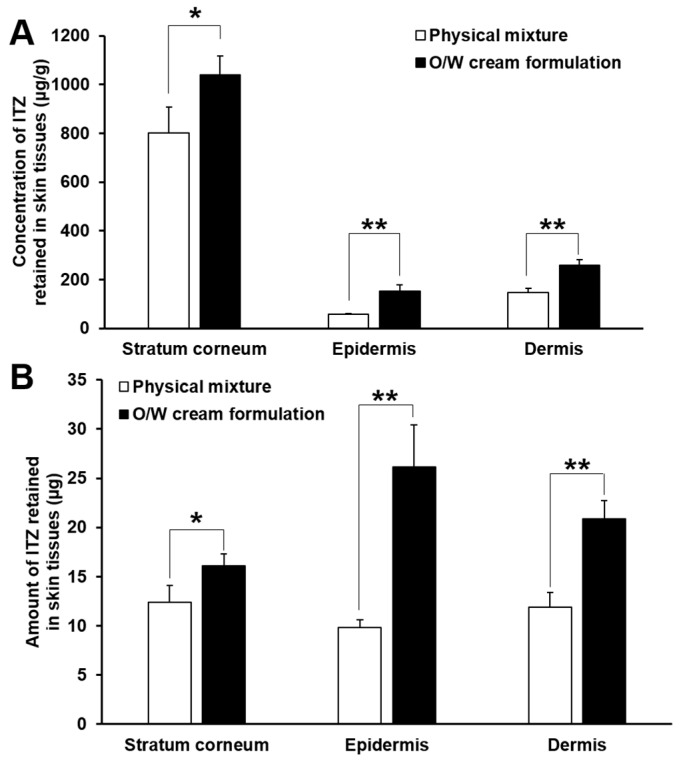
Comparison of concentration (**A**) and amount (**B**) of ITZ deposited in the stratum corneum, epidermis, and dermis evaluated from the physical mixture and O/W cream formulation at 24 h after the start of the in vitro skin deposition study. The values are mean ± SD (*n* = 4). Single and double asterisks indicate statistical differences at *P* < 0.05 and *P* < 0.01, respectively, in the concentration or amount of ITZ deposited in each skin tissue.

**Table 1 pharmaceutics-11-00195-t001:** Solubility of ITZ measured in different excipients such as oils, emulsifiers, glyceryl esters of fatty acid, fatty acid, and fatty alcohols at 20 °C and 80 °C. The solubility of ITZ could not be evaluated in Span^®^ 60, glyceryl mono-stearate, polyglyceryl-3 methylglucose distearate, stearic acid, cetyl alcohol, and cetostearyl alcohol at 20 °C because the excipients were not in a liquid state at 20 °C. The data are presented as the mean ± SD (*n* = 4).

Excipients Tested to Evaluate the Solubility of ITZ		Solubility of ITZ (mg/mL)	
		20 °C	80 °C
Oils *	Labrafac^®^ CC	0.18 ± 0.01	3.62 ± 0.08
	Mineral oil *	Insoluble	Insoluble
	Paraffin oil *	Insoluble	Insoluble
Emulsifiers *	Tween^®^ 80	1.91 ± 0.19	37.53 ± 1.60
	Tween^®^ 40	1.50 ± 0.08	23.35 ± 0.07
	Span^®^ 80	1.03 ± 0.07	23.68 ± 0.58
	Span^®^ 60	Not measurable *	22.08 ± 0.59
Glycerol esters of fatty acid *	Glyceryl mono-stearate	Not measurable *	19.87 ± 1.11
	Polyglyceryl-3 methylglucose distearate	Not measurable *	18.05 ± 1.83
Fatty acid *	Stearic acid *	Not measurable *	14.50 ± 1.22
Fatty alcohols *	Cetyl alcohol *	Not measurable *	10.43 ± 0.63
	Cetostearyl alcohol *	Not measurable *	8.32 ± 0.67
Oil phase of O/W cream formulation **		18.35 ± 0.12	32.62 ± 0.82

* “Not measurable” indicates that the solubility experiment could not be performed because test excipients were not in a liquid state at 20 °C. ** Oil phase of the O/W cream formulation: The composition of the oil phase is presented in Table 2. Labrafac^®^.CC: Caprylic/capric triglyceride. Tween^®^ 40: Polyoxyethylene sorbitan mono-palmitate. Tween^®^ 80: Polyoxyethylene sorbitan mono-oleate. Span^®^ 60: Sorbitan mono-stearate. Span^®^ 80: Sorbitan mono-oleate.

**Table 2 pharmaceutics-11-00195-t002:** Composition of the O/W cream formulation prepared using excipients selected based on the result of the solubility test.

Phase of Cream Formulation	Ingredients	Content (%)
Oil phase	ITZ	1.0
	Labrafac^®^ CC	23.0
	Tween^®^ 80	7.0
	Glyceryl mono-stearate	4.0
	Stearic acid	10.0
	Cetyl alcohol	6.0
Water phase	Propylene glycol	9.0
	Distilled water	40.0

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
