# Peer review of "Characteristics of Skin Deposition of Itraconazole Solubilized in Cream Formulation"

_pharmaceutics, 2019, doi:10.3390/pharmaceutics11040195_

Reviewer 1 Report

From my point of view, this can be published as it is.

Author Response

RESPONSES TO REVIEWERS’ COMMENTS

      First of all, we very much appreciate the reviewers who carefully checked our manuscript. We are deeply indebted to the reviewers for their time and efforts in reviewing the manuscript. We prudently studied the reviewers’ comments and prepared response pages separately to each reviewer.

Reviewer 1

Comments to the author

From my point of view, this can be published as it is.

[Response] We would like to thank you very much for reviewing our manuscript.

Reviewer 2 Report

Dear authors,

All the reviewer's doubts and comments were elucidated. Congratulations on your work.

Author Response

RESPONSES TO REVIEWERS’ COMMENTS

      First of all, we very much appreciate the reviewers who carefully checked our manuscript. We are deeply indebted to the reviewers for their time and efforts in reviewing the manuscript. We prudently studied the reviewers’ comments and prepared response pages separately to each reviewer.

Reviewer 2

Comments to the author

Dear authors,

All the reviewer's doubts and comments were elucidated. Congratulations on your work.

[Response] We would like to thank you very much for reviewing our manuscript.

Reviewer 3 Report

I have noted that the authors have modified the text and included additional material on rat skin. However, there is a difference between skin permeation and deposition. Although the authors cite publications referring to the use of rat skin to predict permeation across human skin, the amounts recovered from rat skin and their relevance for predicting the amounts present in the different layers of human skin remain open to question.

Author Response

RESPONSES TO REVIEWERS’ COMMENTS

      First of all, we very much appreciate the reviewers who carefully checked our manuscript. We are deeply indebted to the reviewers for their time and efforts in reviewing the manuscript. We prudently studied the reviewers’ comments and prepared response pages separately to each reviewer.

Reviewer 3

Comments to the author

I have noted that the authors have modified the text and included additional material on rat skin. However, there is a difference between skin permeation and deposition. Although the authors cite publications referring to the use of rat skin to predict permeation across human skin, the amounts recovered from rat skin and their relevance for predicting the amounts present in the different layers of human skin remain open to question.

[Response] We fully agree with the limitation of using rat skin for predicting the deposition behavior of itraconazole in human as mentioned by the reviewer. For better simulation of the drug distribution in human skin, to date numerous researchers have investigated various skin models such as excised human skin (cadaver skin, excised skin obtained after plastic surgery), porcine skin, snake skin and reconstructed human skin tissues.

We consider that the use of excised human skin is not generally feasible during the initial development of novel topical dosage forms such as our cream formulation. To the best of our knowledge, porcine skin or reconstructed human skin tissues might be the second best to evaluate the deposition behavior of itraconazole in human skin owing to their histological similarity to human skin. However, they also have their own advantages and disadvantages, causing difficulty in selecting proper barrier.

When we conduct another study for assessing the skin distribution or permeation of a topically administered drug in the future, we might compare different skin models to critically discuss the impact of the skin model on the delivery property of a drug.

We once again agree with your concern and surely shall reflect your comment when designing new experiments.

We very much appreciate the reviewer for good comment, question and concern.

This manuscript is a resubmission of an earlier submission. The following is a list of the peer review reports and author responses from that submission.

Round  1

Reviewer 1 Report

This manuscript described the effect of newly developed formulation of ITZ cream to increase antifungal effect of the cream. It contained several interesting findings. Thus, this reviewer recommands the manuscript to publish in Pharmaceutics as a present form.

Reviewer 2 Report

It is an interesting research field, the cutaneous mycoses are difficult to treat and the topical pathway appears the most appropriate. Many drugs are active against the fungi but difficult to formulate in a topical vehicle because of their physic-chemical characteristics. Itraconazole is very active but very little soluble in water, an aspect that negatively influences the preparation of an appropriate formulation and the distribution of the drug in the deeper skin layers representing the action site. The experimental design, however, is trivial on one side and not well set on the other: the authors have to determine the solubility of ITZ in the final emulsion; there is no need to determine the solubility in the individual excipients, this is not useful information for the research. The physical mixture of ITZ and cream base as control formulation is absolutely inappropriate, since ITZ remains un-soluble or partially solubilized simply because the preparation is not done in the right way as it makes no sense to prepare an acidic formulation absolutely not applicable in vivo (the data do not add value to the research).

Reviewer 3 Report

Over all the paper discuss the formulation possibilities of ITZ. The main statement that the O/W Emulsion is the best delivery system to the dermis is clear. The general question how the permeability is effected should be an future project.

In my opion the text would easier to read, if you the define the mixture simple as mixture and do not write “physical mixture of ITZ and O/W cream base”.

114: How does the participants applicate the cream?

126: Which paper do you used?

Table 2: I prefer shorter titles.

279: The W/O Cream is missing. This would be interesting, because I don’t believe that it an emulsion!

313: Is this reported in literature?

Figure 4: The long title is not my style! The main part should be in the methods chapter.

342: wrong order? Dermis and epidermis

Reviewer 4 Report

Dear authors,

Congratulations on your paper.  I was very pleased to read it.

I only have few comments to improve your study:

Line 50: "However, they possess free flowing nature, and thus, the residence time of the drug on the skin is very short". This statement is only about microemulsions and should be adressed to that.

Line 107: "A minimal amount of benzyl alcohol was added..." Please be more specific.

Line 117: "The age of participants ranged from 24 to 30 years". Please write how many participants.

Reviewer 5 Report

The authors describe the development of cream formulations for the delivery of itraconazole into rat skin in vitro and then present the distribution in the stratum corneum, epidermis and dermis. They demonstrate that their cream was able to deliver more itraconazole than an acidic cream and a simple physical mixture of the drug and the cream base.

The experiments are thoroughly described. However, the main question is the relevance of the drug distribution data observed in rat skin and its usefulness for predicting penetration behaviour in human skin. 

For example, rat stratum corneum is < 5 um thick and the entire rat epidermis is 15-20 um in thickness. This is approximately equal to the thickness of human stratum corneum.

In addition, the authors state that approximately 6.5 mg of drug was present in the formulation samples placed in contact with the skin. Given that a 1% cream formulation was used, therefore ~650 mg of formulation was applied to the rat skin surface for upto 24 h.  This raises the question of the integrity of the skin barrier and whether the skin penetration was artificially high.

The authors need to address these points.